# Paper Forager: Supporting the Rapid Exploration of Research Document Collections

Justin Matejka[1α], Tovi Grossman[2µα], and George Fitzmaurice[3α]

[α]Autodesk Research and [µ]University of Toronto

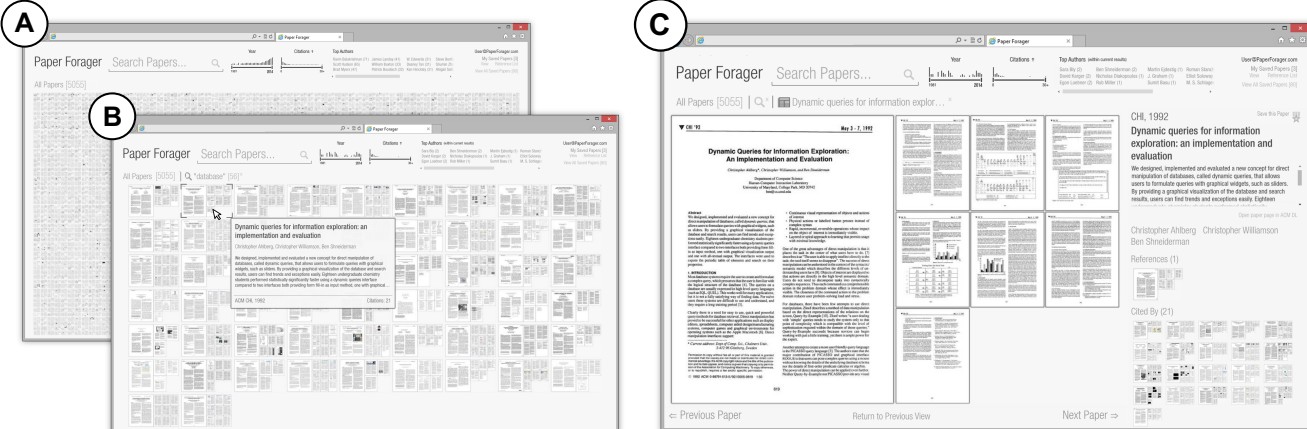

Fig. 1. Three views of the Paper Forager system: (A) the initial state of system showing all 5,055 papers in the sample corpus from the ACM CHI and UIST conferences, (B) the filtered results showing only the papers containing an individual keyword, and (C) a sample paper overview page which further allows a user to click on a page to read the content.

## ABSTRACT

We present Paper Forager, a web-based system which allows users to rapidly explore large collections of research documents. Our sample corpus uses 5,055 papers published at the ACM CHI and UIST conferences. Paper Forager provides a visually based browsing experience, allowing users to identify papers of interest based on their graphical appearance, in addition to providing traditional faceted search techniques. A cloud-based architecture stores the papers as multi-resolution images, giving users immediate access to reading individual pages of a paper, thus reducing the transaction cost between finding, scanning, and reading papers of interest. Initial user feedback sessions elicited positive subjective feedback, while a 24-month external deployment generated in-the-wild usage data which we analyze. Users of the system indicated that they would be enthusiastic to continue having access to the Paper Forager system in the future.

## 1 INTRODUCTION

Literature reviews can be a long and tedious task requiring information seekers to sort through a large number of documents and follow extended chains of related research. With paper proceedings, users can easily *scan* and *read* any of the papers, but *finding* specific papers can be difficult.

In contrast, online digital libraries and search systems improve the ability to *find* specific papers of interest. A number of new systems have been developed [1]–[6] which provide advanced faceted search and filtering capabilities. However, these systems are driven by metadata and textual content and ignore visual qualities such as figures, graphics, layout, and design. Furthermore, such systems

---

[1] *justin.matejka@autodesk.com*

[2] *tovi@dgp.toronto.edu, tovi.grossman@autodesk.com*

[3] *george.fitzmaurice@autodesk.com*

require the user to download the source PDF file before the paper can be read in detail. We seek a single system that can support a continuous transition between *finding*, *scanning*, and *reading* documents within a corpus.

Web technologies such as DeepZoom [7] and Google Maps support browsing of extremely large image-based data sets through the progressive loading of multi-resolution images. This type of architecture is beneficial in that it gives users rapid access to detailed content. However, we are unaware of any prior systems which have used such an architecture for document exploration.

In this paper we present *Paper Forager*, a system to support the rapid filtering and exploration of a collection of research papers. Paper Forager relies on a cloud-based architecture, storing the papers as multi-resolution images that can be progressively downloaded on-demand. By using this architecture, we allow the user to transition from browsing an entire corpus of thousands of papers, to reading any individual page within that corpus, within seconds. In doing so, we accomplish our goal of reducing the transaction cost between finding, scanning, and reading papers of interest.

Our main research contribution is the development of a novel system for literature review, which synthesizes previously explored concepts such as faceted search and zooming based interfaces. We present the design and implementation of Paper Forager and its associated architecture, implemented on a sample corpus of 5,055 papers from the *ACM CHI* and *UIST* conferences. Additionally, we present results gathered from initial user feedback and a 24 month external deployment of the system. Users of the system felt it was easy and enjoyable to use, and the majority indicated that would like to continue using Paper Forager in the future.

## 2 RELATED WORK

### 2.1 Faceted Search

Faceted search allows users to explore a collection by filtering on multiple dimensions. While powerful, representing all of the available options in a user interface can be problematic [8]. Many papers have looked at improving the faceted searching experience. FacetLens [9] represents facets as nested areas on the interface and FacetAtlas [10] displays the relationships between related facets through a weighted

network diagram and colored density map. Pivot Slice [6] used a collection of research papers as a sample corpus, and allows users to explore relationships between facets using direct manipulation. The faceted search system in Paper Forager is designed to be more approachable for new users than the above systems, at the expense of being less versatile in the types of queries which can be performed.

## 2.2 Visual Document Browsing

There have been numerous research projects exploring the space of visually exploring a collection of documents.

The WebBook and Web Forager [11] pre-loaded and rendered web pages so they could be rapidly flipped through, and more recently, Hong *et al*. [12] looked at improving the digital page flipping experience. Document Cards [13] extracts important terms and images from a document and displays them in compact representations.

The DocuBrowse system [14] is designed to browse and search for documents in large online enterprise document collections. Similar to Paper Forager, DocuBrowse includes both a faceted search interface and visual thumbnails of results. While source content can be opened, it is not clear how long it would take to download and view an individual document. Paper Forager expands upon ideas from the DocuBrowse interface, and uses a cloud-based architecture to support rapid viewing through the progressive loading of multi-resolution images. Paper Forager also takes advantage of the connections, such as citation networks, while DocuBrowse supports a wider range of file types without looking at their interconnectivity.

While not directly related to document browsing, the PhotoMesa system [15] allows zooming into a large number of images which are grouped and sorted by available metadata. Similarly, the Pivot Viewer component of the Silverlight framework [16] supports faceted searching of a collection of images based on associated metadata. Results are displayed using a dynamically resizing grid of images, using the Silverlight Deep Zoom technology [7]. We are unaware of attempts to use this type of technology for the exploration of research document collections. Paper Forager implements an architecture similar to Pivot Viewer, but with a customized design and interface for the purpose of rapidly exploring a corpus of research literature.

## 2.3 Research Literature Exploration Tools

There are many deployed systems which provide search access to collections of research papers including Google Scholar [17], Mendelay [18], CiteSeerX [19], Microsoft Academic Search [20], and the ACM Digital Library [21]. For a thorough analysis readers are directed to Gove *et al*.'s evaluation of 14 such systems [4] which highlights the strengths and weaknesses of each system.

There are also research systems which have looked at the topic of research literature exploration. Aris *et al*. [1] and PaperLens [5] are visualization tools which look at paper metadata to show temporal patterns of paper publication, and each uses citation links among papers to explore a field's rate of growth and identify key topics. Along similar lines, the PULP system [22] uses reinforcement learning to find and present a visualization of how the topics in a corpus of research papers have change over time.

GraphTrail [2] is a system for exploring general purpose large networked datasets, and used a corpus of ACM CHI papers as a sample database. GraphTrail supports the piecewise construction of complex queries while keeping a history of the steps taken which allows for easy backtracking and modification of earlier stages. Systems such as Citeology [23] and CiteRivers [24] support exploring scientific literature through their citation networks and patterns, with CiteRivers also including additional data about the document contents. PaperQuest [25] aims to help researchers make efficient decisions about which papers to read next by displaying the minimum amount of relevant information, and considering papers for which the researcher has already displayed an interest.

Another research exploration tool is the Action Science Explorer (ASE) [3], [4]. The ASE system uses a citation network visualization in the center of the interface and makes use of citation sentence extraction, ranking and filtering by network statistics, automatic document clustering and summarization, and reference management.

The main difference between Paper Forager and the above systems is that while these existing systems all perform some amount of analysis, visualization, or filtering based on the metadata or text of a paper, they hide the design, layout, and images of the actual research documents. Furthermore, with existing systems, users must wait until the document is downloaded before reading the paper in detail. Paper Forager provides a basic level of faceted metadata searching along with emphasizing the visual content of the documents, and provides immediate access to reading individual pages of the documents.

An example of a visually-focused research exploration tool is the UIST Archive Explorer [26] which was created for the 20th anniversary of the UIST conference and provided an interface for browsing the collection of papers previously published at UIST. Papers could be viewed by year, keyword, or author. Selecting a paper caused the pages of the paper to be arranged in a row and the user could zoom in for more details. Compared to Paper Forager, the UIST Archive Explorer used a smaller corpus of documents (578 vs. 5055), was hosted locally (whereas Paper Forager uses a cloud-based architecture), and did not allow for navigation between papers based on their citation networks.

## 3 THE LITERATURE REVIEW PROCESS

The theory of information foraging [27] suggests that information seekers try to find documents with potentially high value and then use the available informational "scent" cues to determine which documents, if any, are worthwhile to examine further. We can thus think about the process of literature review being composed of three main stages:

*Finding*: filtering the collection of all possible papers down to those you might want to read, either by browsing the collection, or explicitly searching.

*Scanning*: making a decision for each individual paper as to whether it is worthwhile to read based on the available information scent cues.

*Reading*: looking through the content of the paper for useful information.

In order to maintain flow [28] during the literature review process, it is desirable for the transitions between the stages to be as smooth as possible. Research exploring the dynamics of task switching [29], [30] has shown that small interaction improvements can cause categorical behavior changes that far exceed the benefits of decreased task times.

When papers were primarily distributed in printed proceedings, the *finding* phase of the process was inefficient. However, once a collection of possibly relevant papers was found, the process of *scanning* the papers consisted of flipping through the pages. The informational scene cues [27] presented to the information gatherer to make a reading decision consisted of what was visible in the printed form of the paper – namely the title, text, figures, and the paper's overall graphic design and layout. Based on these cues, a decision to read or not would be made, and the cost of transitioning between the *scanning* and *reading* phases was minimal (Fig. 2).

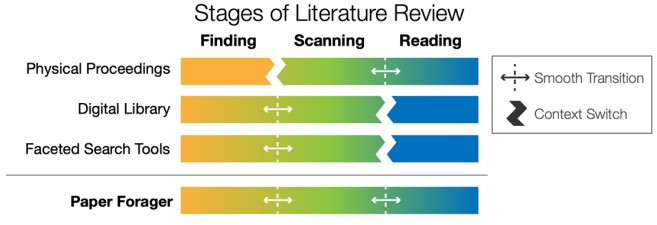

Fig. 2. Four main approaches to paper discovery the context switches required between the various stages of the literature review process.

With digital libraries the *finding* phase of the process is much more efficient, and the transition cost between *finding* and *scanning* was greatly reduced. However, the available informational scent cues presented during the *scanning* phase was reduced to basic textual information such as the title, authors, and sometimes the abstract of the paper.

Advanced paper browsing tools such as ASE [3] provide additional functionality in the *finding* phase as well as incorporating additional scent cues to inform the reading decision such as visualizations and statistical measures of keywords, authorship, and citation networks. But still, the images and visual design of the original paper are not available to the researcher during the *scanning* phase; the graphics of a paper are not visible until after the decision has been made to move from *scanning* to *reading*. Additionally, the transaction cost when deciding to read a paper is relatively high: the paper needs to first be downloaded, which even on a fast network can often take between 3 and 15 seconds, and then it is opened for reading in a secondary application (or at least a new window within the same application). Besides the time cost, the context switch to a secondary application can be disrupt the flow of the information gathering process.

## 3.1 Design Goals

With Paper Forager, we want to take the quick searching and filtering benefits of modern advanced paper discovery systems and combine them with the visual qualities and benefits of paper proceedings. Additionally, we want to reduce the cost of transitioning between stages (Fig. 2) which will improve the flow of the literature review process and encourage a wider exploration of the paper space. By supporting more exploration, the system may put users in a position to make more serendipitous discoveries [31].

## 4 PAPER FORAGER

We created *Paper Forager* to address the problems encountered while exploring large collections of research papers. As a sample corpus we used 5,055 papers published at the ACM CHI and UIST conferences. The metadata was collected using the Microsoft Academic Search API [20] and the source documents were automatically downloaded using links from Google Scholar where possible and manually downloaded from the ACM DL otherwise.

The Paper Forager interface is composed of a set of interface controls at the top of the screen, and a main display area below. On startup, Paper Forager arranges all documents in the collection in the main display area, sorted with the oldest papers at the top and new newest at the bottom (Fig. 1A).

## 4.1 Interface Controls

Along the top of the window are the interface controls for refining the displayed collection of papers which includes the *search field*, *histogram filters*, *author list*, *history bar*, and *saved paper controls* (Fig. 3).

### 4.1.1 Search Field

On the left is the *search field* (Fig. 3) which initializes keyword searches of the titles and abstracts of the papers, as well as searches for authors and conference titles. The search system will automatically recognize author and conference names. For example, a search for "database" would find all papers with the term "database" in the title

or abstract (Fig. 1B), whereas a search for "Buxton" would be recognized as an author search for "William Buxton" and would find all papers published by that author. Additionally, searching for "CHI" or "UIST" will return all papers published at the respective conference, and adding a year to the end of a search term, such as "CHI 2007", modifies the filters to show only the papers from the 2007 edition of the CHI conference.

By default, entering a term in the search field will perform a new query using the entire collection as input, but prefacing a search term with a plus sign (+) creates an additive search filter. For example, if after searching for "Buxton" the user searches for "+mouse", only papers authored by William Buxton which include the term "mouse" will be displayed.

### 4.1.2 Histogram Filters

Beside the search field are *histogram filters* displaying the number of papers published in each year and the relative distribution of the number of citations each paper has received (Fig. 4). Users can click the *Year* and *Citations* headings to set the sorting order of the papers in the main display area. As search events occur, the histograms dynamically update and animate to reflect the distribution for the actively displayed grid of papers.

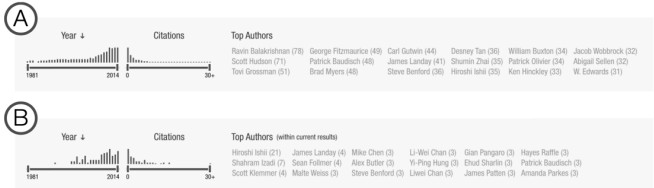

Fig. 4. (A) Histogram filters and Author List for all papers in the CHI and UIST corpus and (B) after searching for the term "tangible".

Under each histogram is a dual value slider which allows the selection of displayed papers to be limited to a specific range of years or number of citations.

### 4.1.3 Author List

To the right of the filter histograms is a list of the top authors of the papers within the current search results (Fig. 3). For example, Fig. 4A shows that Ravin Balakrishnan has the most papers overall in the database, while Fig. 4B shows that Hiroshi Ishii has the most papers for the search term "tangible". Clicking on an author name is equivalent to creating an additive search for the author, so in Fig. 4B, clicking on "Scott Klemmer" is equivalent to entering "+Scott Klemmer" in the search field, and will result in showing all papers for the term "tangible" which have Scott Klemmer as an author.

### 4.1.4 History Bar

Previous research has demonstrated the benefits of keeping a history of actions during information foraging [2], [32]. The *history bar* in Paper Forager is designed for this purpose and provides a way for users to see how they arrived at their current view and the ability to easily backtrack if desired.

Each type of search event has its own history token icon (Fig. 5, A-G) and as the histogram filter sliders are adjusted, the ranges are displayed beside the description of the active search (Fig. 6, H-K). The

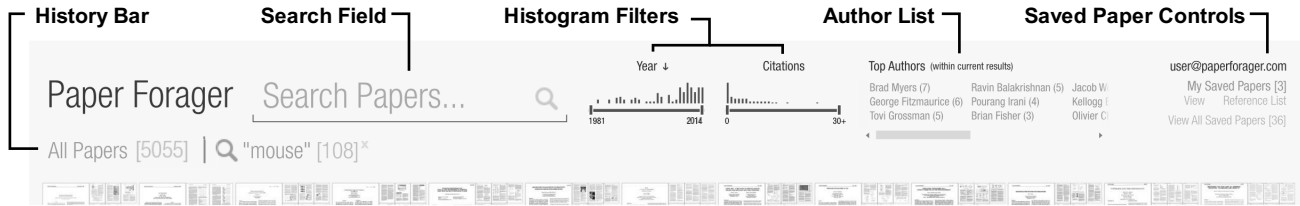

Fig. 3. The interface controls of Paper Forager.

number of results matching the query is displayed in square brackets at the end of the history token.

Each search or filtering event is accompanied by a new token in the history bar (Fig. 7). As the list of tokens grows longer, the previous ones are minimized to show only their icon and their full description is displayed in a tooltip.

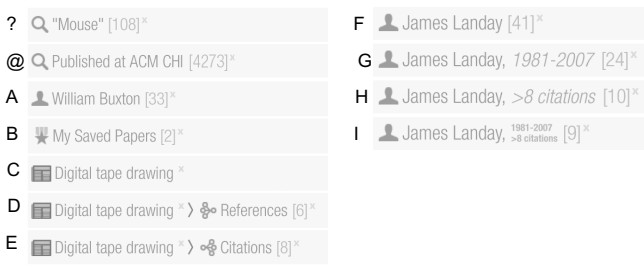

Fig. 5. History tokens for (A) search terms, (B) conferences, (C) authors, (D) saved paper lists, (E) individual papers, (F) references of a paper, (G) citations of a paper, and tokens with filters applied (H-K).

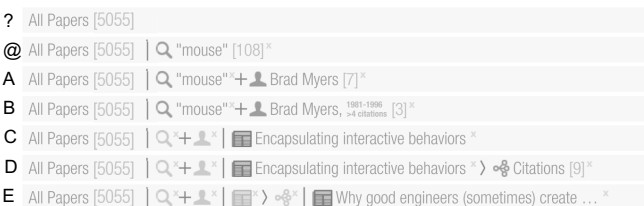

Fig. 6. Initial state of the history bar (A) and changes after a series of operations: (B) searching for "mouse", (C) clicking on the author Brad Myers, (D) adjusting the year and citation filters, (E) selecting a paper, (F) viewing that paper's citations, and (G) selecting another paper.

Inserted between the history tokens are three different separation symbols (Fig. 7): a vertical line when the new state is independent from the previous one, a plus sign when an additive query is entered, and a right facing arrow when looking at references or citations of a particular paper.

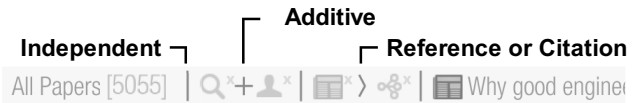

Fig. 7. History token separators.

Clicking on a token in the history list will remove all subsequent query events leaving the clicked token as the active search state. The tokens also include an 'x' button to remove the query from the history list.

### 4.1.5    Saved Paper Controls

Paper Forager allows users to mark papers as saved. The collection of the user's saved papers, as well as all papers saved by the user community can be accessed through links in the top right corner (Fig. 3). Besides accessing the collection of saved papers for viewing, clicking the "Reference List" button copies a formatted list of paper references suitable for a "References" section of a paper to the user's clipboard.

## 4.2    Main Display Area

The main display area offers a *collection view*, a *paper view*, and a *page view*.

### 4.2.1    Collection View

The *collection view* is used to display all papers that match the current query and filters. Papers within the collection are sized so that all results are initially within view. As searches are performed the grid of papers is animated to remove those papers which do not satisfy the query and re-arrange those that do to fill the available space (Fig. 8).

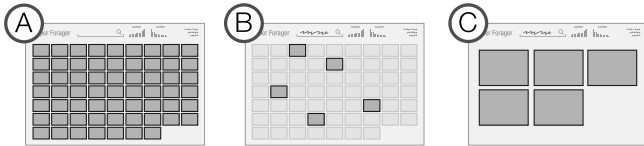

Fig. 8. Stages of the reordering animation. (A) initial state, (B) removed papers fade away, (C) remaining tiles move and resize into new position.

The total animation time is 1.5 seconds, where the outgoing tiles fade out for the first 0.75 of a second, and the remaining tiles rearrange for the next 0.75 seconds. A similar animation occurs when papers not previously on the screen are added.

As the cursor moves around the grid of displayed papers, the paper under the cursor highlights and a large tooltip is displayed with the paper's title, abstract, authors, year, conference, and number of citations (Fig. 9). Clicking on a paper will bring that paper into focus in the *paper view*.

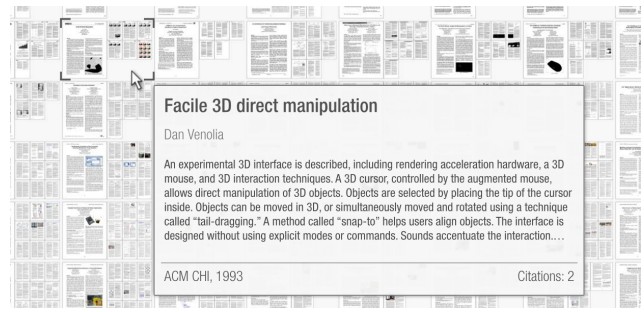

Fig. 9. Example of a paper tooltip.

### 4.2.2    Paper View

Once a paper is selected, either by clicking on a single paper, or by executing a query with only one result, it is displayed in the *paper view* (Fig. 10). Here, the composite image of the paper is fit to the main canvas area, with additional metadata including the title, abstract, authors, venue, and year displayed on the right. A badge icon can be clicked to add the paper to the user's list of saved papers. Clicking an author's name will load all papers by that author (equivalent to searching for the author's name), and there is also a link to follow the DOI link for the paper to view the official page in the ACM Digital Library.

The lower section of the side panel contains thumbnails for each of the papers in the corpus which are referenced by the active paper, as well as all the papers which cite the active paper. Hovering over these thumbnails triggers the associated paper tooltip (Fig. 9) and clicking on a paper thumbnail adds the paper to the history bar and brings it into focus. Clicking on either of the "References" or "Citations" labels takes the system back to the collection view, displaying all of the referenced/cited papers.

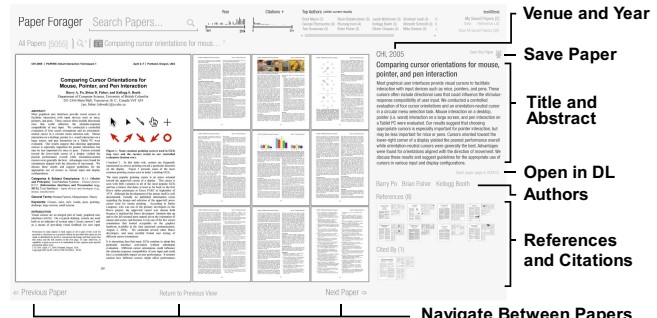

Fig. 10. Interface elements of the single paper view.

Below the paper image is a button to return to the paper collection view, as well as buttons to navigate to the previous and next papers in the current collection. For example, after searching for "mouse" and selecting a paper, repeatedly clicking on "next paper" will let you flip through all papers for the term "mouse". This functionality is also accessible through the left and right arrow keys.

### 4.2.3 Page View

Clicking on a single page animates the display to fit that page into the view (Fig. 11), allowing users to read individual pages. In this *page view*, the navigational controls and arrow keys change to support navigation between the pages of the document.

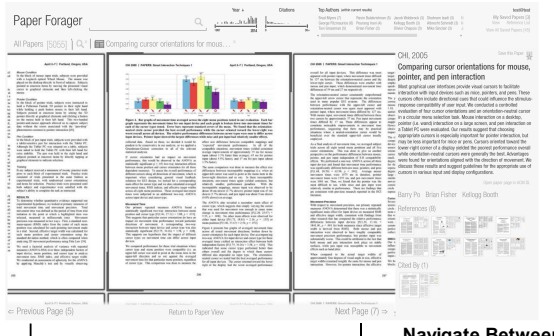

Fig. 11. The page view displays individual pages.

Once the last page in the paper is reached the view zooms back to the *paper view*, and subsequent navigation operations will navigate at the paper level. This enables an efficient workflow of first flipping through papers, then going through the pages of an interesting paper, and then coming back out to flip through more papers (Fig. 12).

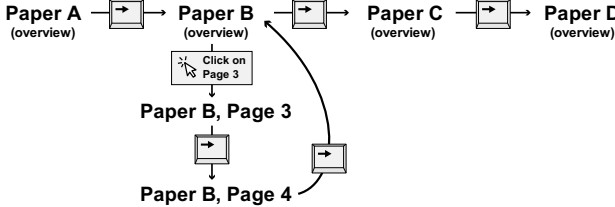

Fig.12. Workflow for navigating between and within papers. (Note: "Paper B" has only 4 pages.)

The layout of the main window is designed so that on 24" or larger monitors the body text of the focused page is large enough to be read comfortably. For smaller monitors, or for more detailed examination of a portion of a page, the *page view* supports zooming and panning with the mouse wheel and left mouse button respectively.

### 4.2.4 Preloading Images

On a reasonably fast broadband internet connection it takes approximately 2 to 3 seconds to download and display a composite paper image (such as in Fig. ) on a 24" monitor. This is an unacceptable delay if trying to rapidly flip through a collection of papers. To address this, when a paper is brought into single *paper view*, the images for the previous and next papers are automatically downloaded and composited at the proper resolution so they can be immediately displayed when requested.

### 4.2.5 Interaction Model

The intent of the Paper Forager design is to support a primary interaction model of searching or filtering for relevant documents, and then clicking on papers or pages to enlarge their view to see them in more detail. Additionally, similar to zooming user interfaces [33], the collection, paper, and page views support interactive zooming and panning. We anticipate that even though the system supports free-

form panning and zooming, that users will prefer, and gravitate towards the search/filter/click interaction model.

### 4.3 System Implementation

Paper Forager is implemented as an in-browser application using the Microsoft Silverlight framework. During development, this allowed for the application to be used in browsers on both Mac OS X and Windows computers with the Silverlight runtime is installed. Due to recent changes to the plug-in architects of major browsers now limit the Silverlight runtime to Internet Explorer on Windows.

The components of the deployed system (Fig. 13) are hosted and stored using parts of the Amazon Web Services (AWS) framework. The application binaries, images, and metadata are stored on and hosted from an AWS Secure Simple Storage (S3) instance. Usage log data and saved paper information are stored in separate AWS SimpleDB (SDB) tables. Due to cross-domain security policies which restrict communication of Silverlight applications, an AWS EC2 server hosts and interprets PHP scripts which facilitate communication between the application and the databases.

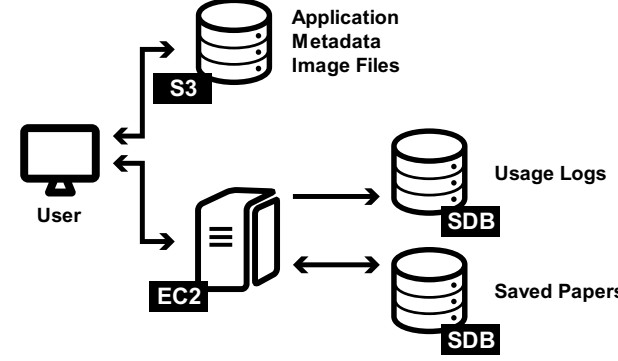

Fig. 13. System architecture diagram.

### 4.3.1 Image Pyramids

To enable fast streaming of papers over the internet and allow the papers to be viewed at a range of resolutions from very small thumbnails up to a large size suitable for reading, papers were converted in to a collection of "image pyramids" following the Microsoft Deep Zoom file format [7]. Each document is rendered at 14 resolutions, from the smallest size of 1 pixel square, up to the original size of the image, in our case, 10,048 pixels wide by 6098 pixels tall. At each resolution of the "pyramid", the images are divided into smaller "tiles" so that only the parts of the image which are needed at that resolution are downloaded (Fig. 14).

We tried maximum tile sizes of 256, 512, and 1024 pixels and found that 512 pixel square tiles provided the best performance for the types of images streamed with our system. On the client side, a Silverlight MultiScaleImage component handles downloading and compositing the tiles to display the image at the requested resolution.

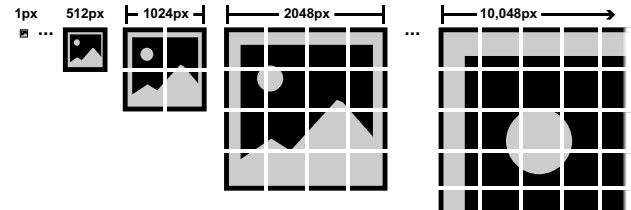

Fig. 14. Image Pyramid data format example.

### 4.3.2 Data Processing

The original PDF versions of the papers go through a multi-stage processing pipeline to convert them into their multi-scale image pyramid format (Fig. 15). First, the PDF files are split into individual pages and converted to JPG image files at 300 dpi. Using the "Data

Sets" feature of Adobe Photoshop, composited PSD files are created combining all the pages of the paper into a single image (Fig. 16). The last step of the process involves converting the large combined JPG image into the image pyramid format.

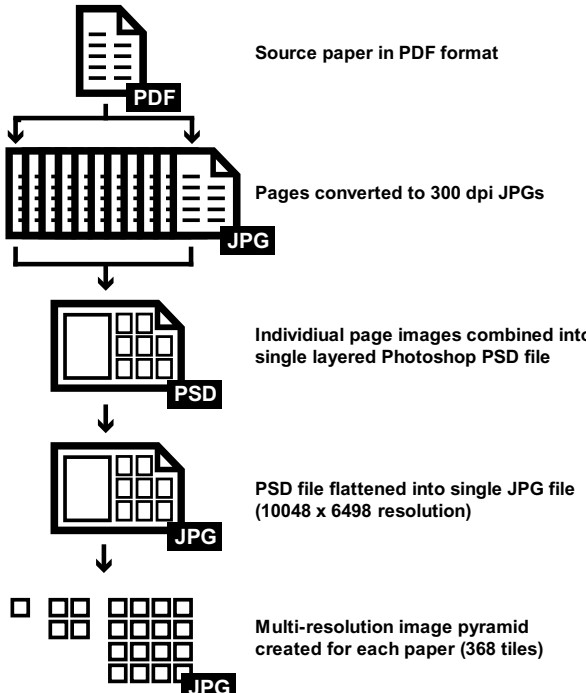

Fig. 15. Data processing pipeline.

The conversion process for each paper took approximately 1 minute on a workstation computer with 24GB of ram and dual 2.53GHz Xeon processors, and the entire sample corpus of 5,055 papers took approximately 90 hours to process, producing ~1.9 million small .jpg images, which generated ~54GB of total image data. Each paper can be processed independently, so the pipeline is well suited for parallelization or computation on remote clusters or servers.

### 4.3.3 Paper Layout

If the paper has 5 or less pages, it uses the 5-page template, and otherwise it uses the 10-page layout. This version of the system did not support papers with more than 10 pages, but it would not be difficult to extend this pattern one more level to a 17-page layout (1 large first page, and a 4-by-4 grid for subsequent pages).

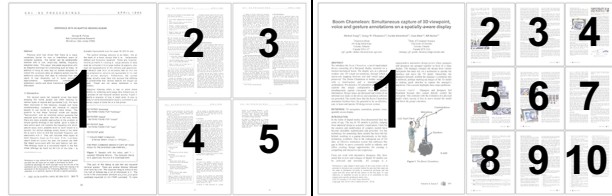

Fig. 16. Sample 5-page (left) and 10-page layouts (right).

We choose to combine all pages of each paper into a single image object before creating the image pyramid as a performance optimization to limit the number of individual objects the system would need to display at any one time. Alternative strategies will be discussed as future work.

## 5 EVALUATION

Quantifying the benefits of information visualization systems is notoriously tricky [34]. To gain insights and usage observations related to our system, we ran two evaluations: a small controlled session to collect initial user feedback, and then a broad, long-term external deployment.

### 5.1 Initial User Feedback

We conducted a qualitative user study to evaluate the features and usability of the Paper Forager system. We wanted to collect initial feedback from users, and validate that some simple (and not so simple) tasks can be accomplished by users in a reasonable amount of time. We recruited 6 participants that were taking an HCI course at a local university (4 male, 2 female, ages 21-24). These students had recently completed a project which required them to gather references for a HCI topic of choice. As such, they were ideal candidates to give feedback on our system and provide a comparative analysis of Paper Forager to the systems and strategies that they had independently used for their literature reviews.

The feedback sessions began with a 5 minute overview demonstrating the main features of the system, after which the participants explored the system on their own for an additional 5 minutes. The sessions concluded with the participants completing a series of 8 tasks, of generally increasing difficulty (Fig. 17).

The tasks were devised such that some could likely be accomplished with a standard digital library search system, some would benefit from faceted searching capabilities, and three of them (c, e, and h) would be prohibitively difficult to accomplish without the added capabilities afforded by the Paper Forager system. The goal of the tasks was to encourage the participants to try different aspects of the system rather than cover all possible use cases for the application. After completing the tasks, participants were asked for thoughts about the system and suggestions for improvements.

### 5.2 Results

All 6 users were able to complete the 8 tasks. While the tasks were not specifically designed to test the speed of using the Paper Forager system compared to traditional digital libraries, task completion times were recorded to see the range of completion times for the various tasks across the set of participants.

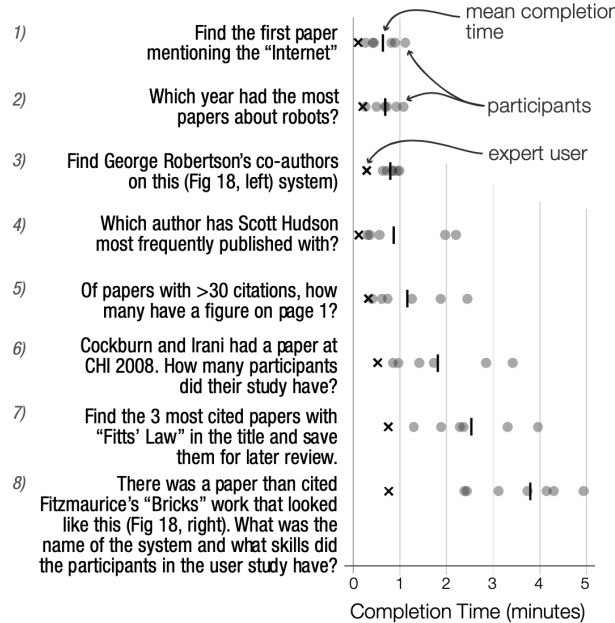

Fig. 17. Task completion times for the 6 study participants, as well as times from one 'expert' user.

Mean task completion times ranged from 33 seconds (task *1*) to 3 minutes and 45 seconds (task *8*). In addition to the 6 study participants, a Paper Forager user with approximately 3 hours of experience was asked to perform the tasks to benchmark expert performance levels of

these tasks. The longer time for the last task was due to participants not always knowing which part of the paper to read in detail to find the necessary information (Fig. 17)

It is interesting to note that for the tasks *1* through *7*, the fastest times from the "novice" study participants after their brief introduction to the system are similar to the completion times from the "expert" user, suggesting that some of the novice users were becoming proficient with using the system after only a short amount of time.

In the comments section of the survey half (3 of 6) of the participants mentioned that their favourite feature was the ability to string together multiple queries with the "+" operator, and 2 of 6 commented that they particularly liked that they could see thumbnails for the referenced and cited papers in the *paper view*. During the 5 minute exploration phase, all participants experimented with the dynamic zooming and panning functionality using the mouse. However, during the tasks, they chose to use the search/filter/click interaction style. Additional features which were requested included auto-completion in the search field, additional conferences in the corpus, and more social sharing capabilities. Overall, participants were extremely enthusiastic about the system, and were hopeful that it would be publically released so they could continue to use it.

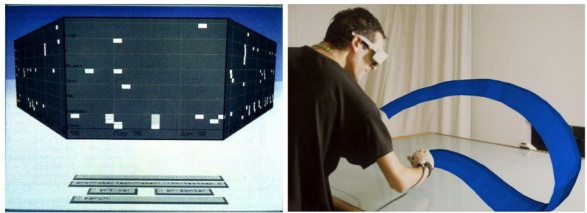

Fig. 18. Images used in questions *3* (left) and *8* (right).

With the overall positive feedback of the system and the confirmation that users of the system would be able to complete some useful tasks, we went forward to go forward with a broader deployment.

### 5.3 External Deployment

To gain additional feedback and in-the-wild usage data, as well as to validate the deployability of our cloud-based architecture, we conducted a long-term external deployment of the system. To maintain compliance with ACM copyright policies (as the papers used in the system are from ACM CHI and ACM UIST), access to Paper Forager was restricted to users with a private ACM account with access permissions for CHI and UIST papers, and to IP ranges with site license to the ACM Digital Library (such as most post-secondary institutions). The system was deployed and available for use continuously over a 2 year period.

### 5.4 Usage Data and Feedback

Over the 24-month deployment period, 493 log-in events were registered from 153 unique users, with 49 of the users logging into the system more than one time. There were a number of "regular" users with 20 users logging into the system more than 5 times each, and 11 users logging more than 100 minutes of active usage. A total of 1,887 papers were viewed in "paper view mode" (Fig. 10) and 1,851 searches were performed over the course of the deployment.

#### 5.4.1 Types of Usage

Since Paper Forager was designed to support the various stages of the literature review process (Finding, Scanning, and Reading), we analysed the log data to see if people were using the system in different ways, or if all users were using the system in a similar manner. To do this we looked at usage along two dimensions: *Browsing vs. Searching*, and *Scanning vs. Reading*.

#### 5.4.2 Browsing vs. Searching (Methods of "Finding")

In this dimension we are looking at different ways users can locate papers which might be relevant, during the *finding* phase of the

literature review process. Using the system in a more "browsing" manner would involve looking at collections of papers, following citation or reference links, and reading many tooltips. Alternatively, a more "search-based" approach to the finding process involves specifically entering search terms into the search field. This dimension is calculated as the user's ratio of "search" events to "browsing" (viewing collections, inspecting tooltips) events.

#### 5.4.3 Scanning vs. Reading

In the *scanning* phase of the literature review process, a user is quickly looking at papers to figure out if they are worth *reading*. In Paper Forager, a reasonable proxy for a user spending lots of time in the *scanning* phase could be a user looking at many papers in the overview *paper view* mode, and zooming into view many single pages in the *page view* could indicate a user spending lots of time in the *reading* phase. This dimension is calculated as the ratio of "paper view" events to "page view" events.

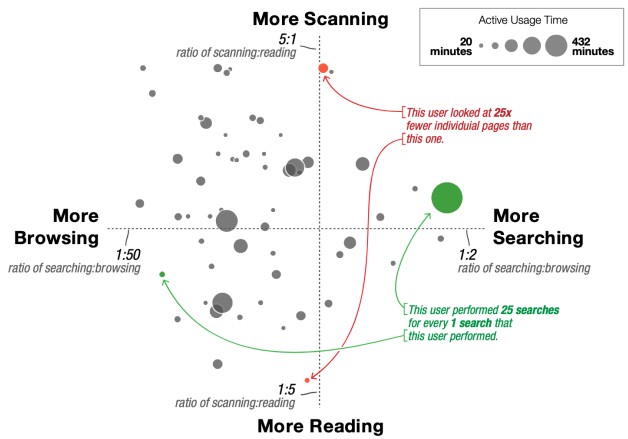

Fig. 19. Usage log analysis showing usage patterns for *finding* behavior (x-axis) and *scanning* vs. *reading* (y-axis).

The 60 users of the system with multiple logins or more than 20 minutes of continuous usage have their activity plotted along these two dimensions in Fig. 19. Each point represents a user, with the size of the point proportional to the amount of activity for that user. Each axis spans a 25x difference in behaviour; that is, users at the bottom of the chart looked at 25x more individual pages than users at the top, and users on the left side performed 25x more searches than those on the right.

It is interesting and encouraging to see that users exhibited such a wide range of usage behaviours. Even among the most active users (those with larger circles) we can see they are distributed around the plot, suggesting that the system can be successfully used for different stages of the review process.

#### 5.4.4 Feedback and Suggestions

At the end of the deployment each user who logged into the system was sent a short voluntary questionnaire (30 of 153 responded, 20% response rate), where they were asked to answer five questions on a 5-point Likert scale (Strongly Disagree, Somewhat Disagree, Neither Agree/Nor Disagree, Somewhat Agree, Strongly Agree):

- *I found Paper Forager **easy** to use.*
- *I found Paper Forager **enjoyable** to use.*
- *Paper Forager is a more **effective** way to research papers than the techniques/systems I have been using previously.*
- *Paper Forager is an **efficient** way to explore research papers.*
- *If kept up to date with papers in my field, I would use Paper Forager to explore research papers in the future.*

The first four questions where based on the criteria outlined by Jeng [35] on which factors contribute to the usability of a digital library system (learnability, satisfaction, effectiveness, and efficiency).

Results are shown in Fig. 20. In general, users felt Paper Forager was easy and enjoyable to use, and a majority of users said if kept up to date with papers in their field, they would continue to use Paper Forager in the future.

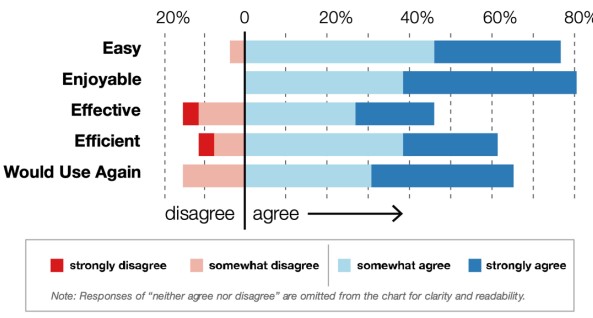

Fig. 20. Results from the subjective questions asked after the external deployment.

Besides the subjective questions, users were also asked to provide details about features they liked and suggestions for improvement.

Several users relayed interesting ways in which they used the system. One Ph.D. student was writing his first paper with a new supervisor and wanted to ensure that his paper followed the general conventions that the supervisor had used in the past. By searching for the supervisor's name and rapidly flipping through his previous papers the student was able to get the answer to a number of questions about the supervisor's style:

> *How many figures does he usually include in a paper? Does he dock figures at the top and bottom of columns, or does he float them in the middle? Does he like using long figure captions? Does he use a particular color scheme for charts? How often does he include an explicit "Contributions" section? How does he typically word his conclusions?*

Before the student had access to Paper Forager he was looking at a single example paper of the supervisor's to try and answer these questions; it was too much work to download and look at all of the supervisor's papers individually. With Paper Forager, each of these tasks took a very short amount of time and effort.

Another user (3D user interface researcher) mentioned they used Paper Forager not only in the process of writing papers, for but other tasks as well:

> *It is so extremely fast and easy to search various topics. You get an idea of what has been in a field, dig for follow up papers (in-depth search) or other related papers (breadth search). I have even used it to find the best reviewers for a paper, or find relevant researchers on any topic (committees, collaborations, etc.) This is just how the digital library should look! This tool has saved me HUGE amounts of time.*

Finally, a grad student finishing up their Ph.D. mentioned that Paper Forager changed the way they approached writing papers:

> *It allowed me to rapidly compare papers to get a sense of structure and style. For instance, when I was writing my own paper, I would quickly look at several examples from related papers to understand what was the typical approach.*

> *The responsiveness also allowed me to view more related papers. With Google Scholar or the ACM DL, it's often several clicks to view papers, and I have to download the PDF first; with Paper Forager I can quickly look at a paper and decide whether it's relevant, so I would actively look at more papers than I would have otherwise.*

A common issue with the system was that it covered too few conferences, and users wanted the collection expanded to cover more of their interests. Several users (particularly those with slower computers and larger monitors) had trouble with performance, finding the interface to not be as responsive as they would have liked.

Many users mentioned liking that the references and citations were prominently displayed in the side panel of the *paper view*, and suggested that the links between related papers could be shown even more emphasized by showing the relationships in the main *collection view*. A number of users said they liked the *collection view* as they often remember papers by their "Fig. 1", however for very large collections (such as the entire 5,055 paper collection shown on launch), some users felt the view was not very helpful, and suggested alternatives such as using a different view of papers when they are displayed very small which could more clearly display relevant information.

## 6 DISCUSSION & FUTURE WORK

The Paper Forager system was designed and optimized to work with collections on the order of 10,000 research documents. It will be interesting to look at how the interaction model should change for much larger collections of papers (an entire digital library for example), as well as how the performance of the system would be affected. Additionally, we would like to explore using the system with other collections of documents with citation networks such as patent applications or court proceedings.

Related to the system performance, Paper Forager combines all pages of each paper into a single image object. It would also be interesting to explore the design opportunities that would arise from storing each page of the paper individually. This would allow for more varied arrangements such as selectively showing only the first page of a paper, arranging the pages of each paper in a row, or highlighting the pages with the most figures. In the time since the system was first developed, Silverlight as a technology has become less-well supported (notably, the Silverlight plug-in will no longer run in the Chrome browser). Re-engineering the system as a HTML5/JavaScript web application would be worthwhile.

To preserve the design and layout work the authors put into creating their papers, we maintained the formatting from the original document. However, we are interested in exploring different representations for the papers when they are at small sizes such as those explored in previous work [26], [36], [37]. It would also be interesting to consider automated approaches for determining good miniaturized representations of research papers and other types of documents.

We would also like to look at ways of annotating the thumbnail images to show aspects of the metadata such as number of citations or which papers have been saved the most often. A coloring technique similar to the one used in AppMap [38] were the thumbnails are shaded based on one variable and sorted by another could lead to interesting discoveries. The searching and filtering capabilities of Paper Forager were purposefully simplified to improve the approachability of the system, but it would be useful to explore combining the visual aspects of Paper Forager with an advanced paper filtering system such as ASE [3], [4] or a visualization of the citation space such as Citeology [23].

Using an image format to display papers has some downsides compared to viewing the actual PDF file, even when the image is at a high resolution. For example, users are unable to select text from a paper in Paper Forager. We believe there is a great potential in a hybrid system where multi-scale images would be used to immediately display the paper while a PDF file loaded in the background. Once a PDF is loaded it could seamlessly replace the multi-image representation.

The ACM paper template contains the guidance *"Please read previous years' proceedings to understand the writing style and conventions that successful authors have used."* We agree that this is a useful task for prospective authors, and hope that Paper Forager could serve as a mechanism to simplify this process.

## 7 CONCLUSION

With Paper Forager we have created a cloud-based system which allows users to rapidly explore a collection of research articles. Our tests of the system produced positive feedback from users who overall agreed that Paper Forager was easy and enjoyable to use while being effective and efficient. We believe our work fills an important gap in existing systems for exploring document collections, allowing users to seamlessly transition between finding, scanning, and reading documents of interest. We hope our work can inspire future research and development in the area.

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
