# OpenReview forum: "Paper Forager: Supporting the Rapid Exploration of Research Document Collections"
_graphicsinterface.org/Graphics_Interface/2021/Conference/Second_Cycle — GI 2021_

### Official Review · Reviewer_ZvgF · 2021-05-03
**Nice interface, lacking evaluation and discussion**

**Rating:** 5
**Confidence:** 4

**Review:**

This paper described a web-based interface to explore large collections of text documents (specifically research papers). Paper Forager allows for users to visually scan through a large corpus of research papers and to also search them using keywords and search terms. The system presents the user with various-sized thumbnail images of papers (based on the number of findings, zoom level, etc.) that preserve its structure and layout of papers and provide an efficient way to scan through them. It also provides several other visualization tools, such as histogram filters and a history bar to provide an overview of search results and contextualize the paper. The system is evaluated by 6 student participants who used it to complete a series of tasks of varying difficulty. Also, findings from a 2-year deployment of the system are presented.

I found the overall system description for Paper Forager (Section 4) interesting and thoughtful. I imagine this system can be useful for scanning a set of documents that are consistently formatted (e.g., ACM CHI, UIST papers) especially for users who are familiar with the conventions of academic paper writing in our discipline. Therefore, I am not surprised to see the positive comments of participants about the system who are all HCI students (or ACM library users in the case of the long-term deployment). It remains to be seen if deploying the system in other less constrained contexts can be as successful. This is not an objection and I believe that developing a system specifically designed for a specialized audience can be a strength. However, providing reflections on the relationship between the consistent format of documents in CHI/UIST and the system’s ability to provide an overview by looking at the overall paper structure (possibly in a limitations section) can strengthen the paper.

I was less impressed with the evaluation section of the paper (Section 5). In particular, more information is needed on how long the participants interacted with the system (average session time), what feedback they had for improvements and which parts of the system they found confusing. In my opinion, quantitatively comparing the performance of 6 novice participants interacting with a system over 5-10 minutes with 1 expert user does not provide enough data for determining system efficiency or ease of use. Initial User Feedback is usually most useful if it points out issues that can be addressed in a new design and these are absent from the paper. The number of participants is, of course, much higher in the external deployment (which is a strength of the paper), and the findings are more interesting (especially the way some users commented on being able to scan papers in a field to see what is a “typical approach.” However, I found the survey question somewhat problematic because including only positive questions leaves the survey results open to acquiescence bias (which needs to be discussed as a limitation).

Another key shortcoming of the paper is that it lacks a discussion section. The short subsection called Discussion at the moment only provides ideas for future changes to the interface rather than synthesizing findings from different parts of the paper and connecting them to existing literature, demonstrating what research questions have been answered. I recommend completely rewriting this section.

Finally, the paper has some minor grammatical mistakes and typos that need to be corrected for a final version.

I want to conclude by reiterating that I find the described interface quite interesting and of relevance to GI’s community. However, the current paper has several important shortcomings that need to be addressed before it is ready for acceptance.

---

### Official Review · Reviewer_UL9e · 2021-05-03
**Well described background and theoretical foundation, and system design but discussion of results needs improvement**

**Rating:** 6
**Confidence:** 3

**Review:**

This paper describes a novel cloud-based tool, Paper Forager, to support the finding, scanning and reading process involved in a literature review. The theoretical foundation of the system is an information foraging model. The authors carried out an initial usability evaluation followed by a longitudinal study. The main findings are the users used a variety of methods to access and organise desired papers from a large database, and enjoyed using the system as it was more efficient than other literature search tools.

I enjoyed reading this paper and found that the authors provided a thorough and well-presented explanation and justification of the system development process. However, the user studies provide limited analysis and discussion of results as follows.

Initial study
- The goal of usability studies is often to find difficulties and problems. What were the concerns, errors, issues from this study?

Longitudinal study
- More demographic information about longitudinal participants is required e.g., ratio of male/female, age of participants, experience with literature review processes, etc.
- The paper states that users on left side of figure 19 look at 25x more searches than those on the right but on the right is says More Searching, I think this should be the other way around.
- It would have been useful to know what specific issues that participants had. An interview, rather than just a questionnaire with simple open ended questions would have allowed more depth.
- Need a discussion about strategies and how people’s behaviour fit the theory information foraging, and compare with other systems or reports in the literature about this behaviour.

Future work
- Use for systematic reviews could be discussed.

Technical
- Figure 1 should not be before the Abstract
- Verb tense mismatch in a number of paragraphs (usually past tense is mixed with present tense in the same paragraph – this makes reading more difficult).
- Interface controls section there is (Fig. ) with no Figure reference
- Punctuation is always included within the quotation marks, not after (e.g., “tangible”. Should be “tangible.”)
- There is a reference to a Fig. B, but there is no Fig. B. Should this be 4B? This occurs in other sections (e.g., Preloading Images)
- Section 5.1 – Initial is misspelled.
- There a numerous awkward sentences throughout paper (e.g., Even among the most active users (those with larger circle) we can they…. (this sentences is missing words))

---

### Official Review · Reviewer_nY4p · 2021-05-04
**nice design for supporting the navigation of document collection that's driven by some good insights of the underlying process.**

**Rating:** 7
**Confidence:** 4

**Review:**

The paper presents Paper Forager, a web-based system for supporting users to quickly explore large collections of documents. The current prototype and evaluation target specifically at research papers published at the ACM CHI/UIST conferences. The paper first identifies a behavioral model consisting of different main stages of literature review based on the theory of information foraging. The paper then presents the design goals, designs of interface elements and views, as well as how the system was implemented using a cloud-based system architecture as well as the multi-resolution approach for displaying the paper collection in different views and modes. The evaluation consists of both a small sample testing with predefined tasks, and a field-based longitudinal deployment of the system online for 24-month period of time. Results of task performance (completion time, in study 1), usage log, subjective usability and open-ended feedback were presented in the paper.

Overall, I found this paper well-motivated and enjoyable to read. This is a system paper that describes a "hybrid" approach that aims to integrate text search and visual browsing to meet the needs to explore collection of documents for purposes like literature review. It is important and insightful that the underlying workflow for literature review is identified and categorized as three different yet interconnected phases (finding, scanning, reading), which really helps to motivate the need to have a new system that combine text search and visual browsing, and justifies the general design of Paper Forager.

I also found the interface design and implementation of Paper Forager of very good quality. I can easily understand what have been done in the system. But still I felt that not all design decisions of interface components or system features were fully explained. For example, it would be helpful to explain why histogram filters are included and designed in this way. The design of history bar is also an interesting one, and I wonder how does it fit into different phases of literature survey. Nevertheless, the completeness of system is impressive to me, and is definitely a strength of this work. The novelty of the design isn't necessarily on the visualization or the technical aspect of the system, but mostly on how it embodies and addresses the needs of literature review by enabling diverse searching and visual browsing activities to happen through a single integrated system.

The evaluation, while isn't probably using the strongest method, has provided some good empirical support and interesting findings on how users may interact with the system to accomplish tasks of literature review and document exploration. My main reservations are: (1) the usage log analysis presented in study 2 should be further developed and discussed to make a connection with information foraging. Simply saying that diverse behaviors were observed in a 2-year deployment study doesn't seem to provide much useful insight. (2) Compared to the usability survey, the open feedback provided by users appear to be have more surprising and thus interesting findings. I would recommend to further code the feedback qualitatively to identify themes in light of the model of information foraging. The current discussion is brief, focusing only on system issues. The results of the two user studies should be further discussed in the discussion.

Overall, I would be happy to recommend accepting this paper even though it is not perfect. The approach of integrating searching and visual browsing for literature review appears novel and useful to me.

---

### Meta-Review · Area_Chair_NQXV · 2021-05-06

**Recommendation:** Accept
**Confidence:** 4

**Metareview:**

The reviewers have all provided substantial, concrete reviews on the paper. Overall, the reviewers are in agreement of the novelty and usefulness of the proposed system design to explore a large collection of documents. Reviewers also found the paper enjoyable to read and the system design/development reasonably explained and justified.

Reviewers also identified some weakness and issues of the paper. There's the agreement that the evaluation still can be stronger. While the mixed-method approach consisting of both usability study and longitudinal study provides a good starting point to evaluate the system, reviewers identified several aspects of the studies to improve, such better connections with the theory used to motivate the design, better documentation of concerns, errors and issues in the usability study, better identification of users' subjective issues and concerns through feedback and interviews, and better integration and synthesis of study findings etc.

While the evaluation is not the strongest, the reviewers overall still provide very positive comments and ratings on the paper. So I'm confident in recommending the paper for acceptance, but would also urge the authors to revise the paper, especially the discussion and limitation of the studies, when preparing the final version.

---

### Decision · Program_Chairs · 2021-05-08

Accept